# Dielectric Spectroscopy Studies of Conformational Relaxation Dynamics in Molecular Glass-Forming Liquids

**DOI:** 10.3390/ijms242417189

**Published:** 2023-12-06

**Authors:** Michela Romanini, Roberto Macovez, Sofia Valenti, Wahi Noor, Josep Lluís Tamarit

**Affiliations:** Grup de Caracterització de Materials, Departament de Física and Barcelona Research Center in Multiscale Science and Engineering, Universitat Politècnica de Catalunya, Barcelona East School of Engineering (EEBE), Av. Eduard Maristany 10-14, E-08019 Barcelona, Spain; michela.romanini@upc.edu (M.R.); roberto.macovez@upc.edu (R.M.); sofia.valenti@upc.edu (S.V.); wahi.noor@upc.edu (W.N.)

**Keywords:** molecular conformers, secondary relaxations, activation barrier, characteristic relaxation time, internal rotation, ring buckling, ring inversion, rotational isomerism

## Abstract

We review experimental results obtained with broadband dielectric spectroscopy concerning the relaxation times and activation energies of intramolecular conformational relaxation processes in small-molecule glass-formers. Such processes are due to the interconversion between different conformers of relatively flexible molecules, and generally involve conformational changes of flexible chain or ring moieties, or else the rigid rotation of planar groups, such as conjugated phenyl rings. Comparative analysis of molecules possessing the same (type of) functional group is carried out in order to test the possibility of assigning the dynamic conformational isomerism of given families of organic compounds to the motion of specific molecular subunits. These range from terminal halomethyl and acetyl/acetoxy groups to both rigid and flexible ring structures, such as the planar halobenzene cycles or the buckled saccharide and diazepine rings. A short section on polyesters provides a generalisation of these findings to synthetic macromolecules.

## 1. Introduction

Glasses are often obtained by deep-supercooling liquids fast enough that transformation to the equilibrium crystalline phase is avoided, and “arrested liquids” with solid-like mechanical properties are obtained. Such amorphous solids are nonequilibrium systems that are constantly relaxing, through a process known as *ageing*, toward a denser state that corresponds to the extrapolation of the metastable supercooled liquid to the temperature and pressure conditions of the glass specimen [1].

A fundamental microscopic understanding of the properties of amorphous solids, in particular of their glass transition phenomenology and ageing behaviour, requires knowledge of the temperature dependence of microscopic relaxation processes. From a microscopic perspective, the glass transition of supercooled organic molecular liquids is due to the dramatic slowing down, upon decreasing the temperature, of the so-called structural α relaxation, a cooperative relaxation process which, in small-molecule glass-formers, is directly linked to the viscosity. In this context, the glass transition temperature *T*_g_ is defined as the temperature at which the α relaxation attains a characteristic time of 100 s [2]. Virtually all glass-formers also display, apart from the structural relaxation, a second relaxation process, more local in character and faster in dynamics, known as the Johari–Goldstein (JG) relaxation, whose ubiquitous nature has led to the suggestion that it is a fundamental ingredient of the glass transition phenomenology [3]. The JG relaxation is observed also for rigid molecules and under nanometric confinement, which shows that it is due to the rigid rototranslational motion of one or few molecular units. Its relaxation time can be predicted from the distribution of relaxation times of the α relaxation via the coupling model equation [4,5], which indicates that the structural relaxation process is the collective, many-particle analogue of the JG relaxation [6]. Ageing processes in the glass state are due to either one of these relaxation modes depending on the temperature.

Considering that the glass transition remains an unsolved problem in condensed matter physics, it is no surprise that experimental studies of organic glass-formers have focused mainly on the characterisation of the structural and JG relaxations, due also to their role in determining the ageing of amorphous solids. This focus is also justified by the fact that these two processes are the predominant features observable by a number of spectroscopic techniques, such as dynamic mechanical analysis, dynamic light scattering, or dielectric spectroscopy.

In the case of dielectric spectroscopy, intramolecular relaxation processes are also observed in organic glass-formers, usually at shorter relaxation times and with lower intensity compared to the JG process [7]. These intramolecular processes, which are mainly observed in the glass state in the typical frequency range of dielectric spectroscopy, are due to the interconversion between distinct or equivalent molecular conformers of (at least partially) flexible molecules. Dielectric spectroscopy measures the polarisation induced by the application of an AC electric field, so that only conformational relaxations that involve a change in the molecular dipole moment can be detected. Intramolecular conformational dynamic processes were the focus of several studies in the first years after a reasonably large frequency domain started to be available for dielectric spectrometers. This initial interest in the dynamic stereochemical information that could be extracted from dielectric studies is testified, for example, by the chapters devoted to intramolecular relaxations in the monograph by Böttcher and Bordewijk [2].

However, many of these initial studies ignored the existence of the JG relaxation and were carried out in limited frequency ranges or at only few frequencies, so that the spectral information they provided could not be reliably assigned to specific relaxations. Another common limitation was the limited temperature range available in most experimental set-ups, so that many results were obtained near room temperature in the normal liquid phase of the studied compounds. In such cases, the secondary relaxations were not clearly spectrally resolved from the structural relaxation. The strategy commonly employed to identify and study the secondary processes of a given compound was to dissolve it in a viscous nonpolar solvent, such as benzene, xylene, or o-terphenyl [8], or achieving dispersions of the compound in nonpolar polymers, such as polystyrene, polypropylene, and polyethylene [9]. This strategy must also usually be employed in NMR studies of conformational relaxations [10]. Already, these initial studies indicated that the intramolecular conformational relaxation processes of a given molecule displayed similar characteristic times and activation energies, regardless of the nature of the samples (pure compounds, solutions, polymer dispersions), implying that they are largely unaffected by intermolecular interactions (with the exception of hydrogen-bonded systems, as discussed below).

In recent years, several temperature-dependent dielectric spectroscopy studies have provided so-called relaxation maps of all dielectrically active molecular processes (namely, those that involve a change in the molecular dipole moment between the initial and final conformations, or during the relaxation via an intermediate conformer) in the condensed phase of suitable pure compounds, in relatively wide temperature intervals covering their liquid, supercooled liquid, and the glassy state. In such studies, the identification of the α and JG processes has become almost a routine procedure, and the intramolecular dynamics can be relatively easily identified as such. The advantage of dielectric spectroscopy experiments is that they provide direct information on the relaxation times and activation energies of the involved processes; their drawback is that the exact assignment of these interconformer conversion processes is unavailable due to the lack of chemical information. Nonetheless, as it was noted early on [11], a survey of literature data suggests that, as a rule of thumb, the characteristic relaxation time of intramolecular processes at a given temperature increases with increasing size (mass, moment of inertia) of the moiety involved in the conformational interconversion. At the same time, the activation energy increases with size. It is therefore possible, in principle, to distinguish between internal rotations of small or large submolecular units, although it should be noted that rotational barriers depend also on the distribution of electronic orbitals in the equilibrium and intermediate-state conformations, which is independent of the mass.

As reviewed herein, a comparison between compounds with a similar molecular structure (and only few functional groups) and/or correlation with the results available from other experimental or computational techniques can allow a reliable assignment of intramolecular dielectric relaxation to rotations of specific functional groups or ring moieties. The spectral identification, in dielectric experiments, of relaxation modes associated with specific conformational processes allows gathering detailed information about their characteristic times and activation energies. Such knowledge is more easily extracted from dielectric spectroscopy than from other techniques providing dynamic stereochemical information, such as time- and temperature-dependent NMR spectroscopy; moreover, dielectric spectroscopy allows for the study of intramolecular relaxations in condensed phases of pure compounds.

Knowledge of conformational dynamics in condensed matter is important to understand the possible phases displayed by pure compounds. For example, elongated molecules form nematic liquid crystalline phases with uniaxial properties because the activation of rotational relaxations about their short axes confers them an effective cylindrical symmetry for what concerns excluded-volume interactions. Similarly, succinonitrile, a compound which does not have an obvious globular shape, displays an f.c.c. plastic crystalline phase at temperatures intermediate between the fully ordered crystal and the liquid phase; the formation of such a phase is due to the fact that the interconversion dynamics between distinct conformers leads to effective nondirectional intermolecular interactions and a relatively low density (see Refs. [12,13] and references therein), so that the molecular packing is the same as that for hard spheres. In the case of amorphous solids (glasses), knowledge of secondary relaxations is crucial to establish the temperature range of kinetic stability of the glass, which is normally observed below the temperature at which secondary relaxations become activated [14], and can be important to understand the crystallisation tendency and ageing behaviour below the glass transition temperature. Last, but not least, from the computational point of view, knowledge of conformational relaxation times and activation energies in the condensed phase may be helpful to test the validity of the torsional force fields employed in molecular dynamics simulations with flexible molecules [15,16,17,18,19,20].

Hydrogen-bonding (HB) interactions [21,22] may hinder or slow down intramolecular relaxations; at the same time, they may lead to time-fluctuating HB clusters [23] that can contribute specific dielectric relaxations, such as the Debye relaxation of monoalcohols [24] or the water-induced relaxations in hydrated systems [25]. For example, no intramolecular relaxation is reported in dielectric spectroscopy studies of several alcohols, including 1-propanol [26,27], phenylpropanol [28], and 5-methyl-2-hexanol [27,29,30]. In other cases, the overlap of the Debye and alpha relaxations does not allow an unambiguous assignment of the secondary relaxation, nor to exclude that it is of the JG type [31]. For these reasons, we mainly restrict our review to compounds that do not exhibit (extensive) HB interactions. In such cases, it turns out that the characteristic time and activation energy of intramolecular processes are single-molecule features that do not depend on the local molecular environment; for example, activation energies for the conformational dynamics of a given compound are the same in the gas phase, dilute solutions, and supercooled liquid and glass states of the pure compound. These results, to be reviewed in Section 3, are in agreement with molecular dynamics simulations, which generally show that the conformational energetics are different in the gas and liquid phases of self-associating compounds (e.g., alcohols), while no such difference is observed in nonpolar compounds and in compounds that do not self-associate [32].

In the following, we first summarise the method employed in most dielectric relaxation studies to extract quantitative information about conformational relaxation dynamics (Section 2), and then review (Section 3) some of the available dielectric spectroscopy data on flexible molecular glass-formers, and on two classes of polymeric glass-formers.

## 2. Method

All the dielectric spectroscopy measurements reviewed herein were acquired by different research groups on pure compounds in the supercooled liquid and glass phases obtained by fast-quenching from the normal liquid state above the melting temperature, except in the case of binary mixtures of dinitriles (Section 3.1), where both supercooled liquid and plastic crystalline phases were considered, and in the case of acetylbenzene derivatives (Section 3.2), which were characterised in p-xylene solutions and dispersions in polystyrene. In some cases, data are compared to those obtained by other techniques (NMR, ab initio calculations) either in the gas phase or dilute solutions of the same compound.

Dielectric spectroscopy measures the frequency-dependent complex relative permittivity $\varepsilon^{*}\left(\omega \right)=\varepsilon^{\prime }\left(\omega \right)-i\varepsilon^{\prime \prime }\left(\omega \right)$ of a sample, also known as the *complex dielectric function*, where ω = 2π*f* is the angular frequency of the applied AC field. Isothermal dielectric spectra are usually acquired at ambient pressure at different temperatures, although measurements can also be carried out under an applied hydrostatic pressure. Dielectric relaxations are visible as successive step-like decreases in the real part *ε*′(*ω*) of the complex dielectric function, and more clearly as separate peaks or overlapping spectral components in the imaginary part *ε*″(*ω*), known as the *dielectric loss* function (see Figure 1). The typical interval of frequencies *f* covered in dielectric spectroscopy experiments, where the AC voltage is applied directly on a parallel-plate capacitor, ranges from 10^−2^ Hz up to 10^6^ Hz. Higher frequencies can be reached with such experimental setup, but the loss signal is then dominated by an intrinsic background which introduces significant experimental errors; the upper frequency limit can instead be extended to higher values (e.g., 10^9^ Hz) by making use of waveguides or employing other quasioptical methods [33]. Unless stated otherwise, the data reviewed in this contribution have been acquired either in the low-frequency AC capacitor setup (data up to 10^6^ Hz), or by a combination of the low-frequency AC technique with a waveguide technique (data up to 10^9^ Hz).

The contribution of an intramolecular relaxation (as well as of a JG relaxation) to the complex relative permittivity is usually fitted with a Cole–Cole function [34,35], whose analytical expression is:(1)$${\varepsilon_{CC}}^{*}\left(\omega \right)=\frac{∆\varepsilon }{1+{\left(i\omega \tau \right)}^{c}}$$

Here, Δ*ε* is the dielectric strength (intensity) of the relaxation, *τ* is the characteristic time of the relaxation, corresponding to the maximum dielectric loss at the probed temperature, and *c* is the Cole–Cole exponent that describes the width of the relaxation feature in the frequency domain. It should be noted that a Cole–Cole function entails that the corresponding spectral feature is actually characterised by a distribution of relaxation times, rather than a single one [2]; the physical meaning of the characteristic time τ is therefore that of the average, or most probable, relaxation time for a given conformational relaxation.

Figure 1 shows the frequency-dependent dielectric loss spectra of two molecular glasses, namely, those obtained by supercooling liquid *m*-fluoroaniline (upper panel [23]) and *o*-bromobenzophenone (lower panel [36]) to below their respective the glass transition temperature (*T*_g_). In dielectric spectroscopy, the loss features associated with intramolecular relaxations are usually observed near or below *T*_g_. In some cases, conformational relaxations, usually labelled as γ, are observed as a local maximum (peak) in the loss spectrum, as is the case of *m*-fluoroaniline [23] in the upper panel of Figure 1, while for other compounds, the γ loss feature is observed only as a shoulder to the structural (α) or the Johari–Goldstein (β) spectral components, as for the case of *o*-bromobenzophenone (lower panel of Figure 1), where a double-logarithmic representation of the spectra must be used to observe both relaxation components. The fit components and overall fit quality can be discerned in both panels of Figure 1.

Usually, the characteristic time of intramolecular relaxations has a simply activated Arrhenius dependence on temperature, given by:(2)$$\tau =\tau_{\infty }exp\left(\frac{E_{a}}{RT}\right)$$

Here, *R* is the gas constant, $\tau_{\infty }$ is the relaxation time in the limit of very high temperature, which can be interpreted as the inverse of the “attempt rate” with which the molecule tends to relax, and *E_a_* is the molar activation energy of the relaxation process (so that the “success rate” 1/τ is proportional to the Boltzmann factor $exp\left(-\frac{E_{a}}{RT}\right)$ representing the probability that the relaxation barrier is overcome). Intramolecular (interconformer) relaxation processes are thus characterised by two parameters, Δ*ε* and *c*, which are only weakly dependent on temperature, and by two constants, $\tau_{\infty }$ and *E_a_*, which are characteristic of the given relaxation process.

In view of Equation (2), it is customary to visualise relaxation times as an Arrhenius plot (i.e., as Log(τ) vs. 1/*T*), from which the activation energy *E_a_* can be directly extracted from the slope of the linear fit to the experimental points. It is worth pointing out that earlier studies employed a slightly different version of Equation (2) based on the Eyring–Kauzman relation: $Ln\left(T\tau \right)=\frac{h}{k_{B}}-\frac{∆S_{a}}{R}+\frac{∆H_{a}}{RT}$, and in which, instead of an activation energy, separate activation enthalpy and entropy contributions were considered. This was customary both in dielectric spectroscopy [37,38] and NMR [10] data analysis. Nowadays, relaxation processes are no longer interpreted within such a framework. Data reviewed from articles where this analysis was carried out are reported here as Arrhenius plots with the relaxation time on the vertical scale (and not as relaxation time multiplied by temperature, as in the Eyring–Kauzman analysis), and the activation energy is then extracted from the corresponding slope (Equation (2)).

As mentioned in Section 1, in order to be observed by dielectric spectroscopy, a conformational relaxation must involve a change in the direction or magnitude of the molecular dipole moment; hence, the organic compound under scrutiny must contain at least a functional group containing atoms of different electronegativity, and be at least partially flexible. Apart from unsubstituted hydrocarbons and a few symmetrically substituted ones, all organic molecules fulfil this constraint. On the other hand, the lack of chemical sensitivity of the experimental technique implies that only molecules with a restricted number of well-defined conformers can be successfully analysed. In fact, large molecules with several distinct polar functional groups might display distinct intramolecular relaxations that are partially overlapping in frequency; moreover, molecules possessing functional groups at the end of long aliphatic chains are characterised by several torsional conformers, and thus do not display a single, well-defined relaxation time. This restriction has partially motivated and guided our choice of experimental results to be included in this review.

## 3. Results

This review is structured in consecutive subsections focusing on different types of functional groups that can contribute to the dynamic conformational isomerism of organic compounds. We start from smaller functional groups (terminal halomethyl, nitrile, acetyl, and acetoxy groups) and move to rigid or flexible ring groups (aromatic halobenzenes, saccharide rings, benzodiazepines, and thiacrowns), ending with a short subsection on polyesters.

### 3.1. Haloalkanes, Acetylalkanes, and Simple Nitriles

The basic terminal hydrocarbon group is the methyl (-CH_3_) group. It is well known that methyl group rotation is activated at room temperature in organic compounds, even in the crystalline state. Such rotation is too fast to be observed in the usual frequency–temperature range of dielectric spectroscopy; moreover, it has a very weakly associated dipole moment, so that it would be silent in a high-frequency dielectric experiment. Substitution of the methyl hydrogens with more electronegative atoms, such as halogens, provides a dipole moment that enhances detection by dielectric spectroscopy. The rotation of fluorinated or chlorinated methyl groups, given the small molecular mass of F and Cl, is still too fast to be observed by normal dielectric experiments [39,40]; however, substitution with heavier halogen elements, such as bromine or iodine, leads to slower rotational dynamics that can be detected. The same is true when the substituent is a larger polar group, such as the acetyl group.

Figure 2a shows the Arrhenius plot of the conformational relaxation times of three butane derivatives, namely, butyl iodide (1-iodo butane), butyl bromide (1-bromo butane), and butyl acetate, in their supercooled liquid and glass states. The first two compounds are obtained from butane by the single substitution of one halogen of a high atomic number (I, Br) at a terminal methyl group [11,41]. The intramolecular relaxation of all three compounds is observed only at relatively low temperature, namely, below 155 K, and has a reported molar activation energy equal to 20.1, 20.4, and 21.4 kJ mol^−1^ for butyl iodide, bromide, and acetate, respectively; that is, almost independent of the terminal group. The molecular process associated with the relaxation of both butyl halides could be the internal rotation of the -CH_2_X group, or else involve the rotation of the longer -CH_2_CH_2_X moiety (where X = I, Br) around its covalent bond (or be a combination of both). It is interesting to note that, as might be expected, the relaxation times are longer (i.e., the relaxation dynamics is slower) for the heavier iodine substituent (atomic weight ≈ 127) than for bromide (atomic weight ≈ 80). Similar relaxation times and activation energies are reported for the slightly longer-chain 1-bromo hexane compound (not shown in Figure 2a) [42]. In comparison, the acetoxy terminal group of butyl acetate has a lower mass but more complex chemical structure and distribution of dipole moments, which might slow down the conformational relaxation process.

Another simple modification of the terminal methyl group is by the substitution of a hydrogen by a hydroxyl (-OH) or amine (-NH_2_) group, leading to a monoalcohol or a primary amine. Alcohols are characterised by extensive HB networking, which, as mentioned in Section 1, leads to specific low-frequency relaxation processes in the dielectric spectra (especially for monoalcohols [24]). The rotation of the -OH group around its connecting bond is accompanied by a change in the dipole moment due to the spatial reorientation of the lone electron pairs of the oxygen atom. Such rotation is extremely fast in the gas phase, but can be experimentally accessible in dielectric spectroscopy studies of amorphous condensed phases due to the dynamic slow-down induced by self-association, since the rotation of the -OH group is, in this case, accompanied by the breaking and reformation of the hydrogen bonds [43]. The same is probably true for the sulphonamide group, whose rotation may be responsible for the fastest (δ) relaxation observed in the active pharmaceutical ingredient celecoxib [44]. Due to the intrinsic complexity of the HB dynamics, which may take place in disordered condensed phases of alcohol (-OH), carboxylic acid (-(C=O)OH), and amine (-NH_x_) derivatives, we refrain from including these compounds in the present review, and focus mainly on nitrogenated or oxygenated compounds that are incapable of HB formation with themselves. It is worth stressing that, as also mentioned in Section 1, no conformational relaxation has been observed in dielectric spectroscopy studies of simple alcohols, such as 1-propanol, phenylpropanol, or 5-methyl-2-hexanol [26,27,28,29,30].

In aldehydes and acetyl compounds (such as butyl acetate), the dipole moment stems from a functional group containing a double (carbonyl) bond. Another possibility of a non-HB polar functional group possessing a multiple bond is the terminal nitrile triple bond (-C≡N). Interesting examples of such cyanide compounds are the short-chain dinitriles, such as succinonitrile (SN) and glutaronitrile (GN). Binary liquid mixtures of both compounds at a high GN content can be supercooled from high temperatures to yield a structural glass [12], which allows probing their intramolecular dynamics in a wide temperature range (the conformational dynamics is, in fact, well separated in frequency from the JG relaxation in the mixtures [45]). Moreover, at room temperature, the thermodynamically stable form of pure SN and of SN–GN mixtures is a plastic crystalline f.c.c. phase in which molecules undergo tumbling reorientations at fixed lattice positions; this phase can also be supercooled, leading to the formation of a so-called “orientational glass” [12,46].

As shown in Figure 2b, the intramolecular relaxation of dinitriles, which can be ascribed to the relative reorientation of -CH_2_C≡N or -CH_2_CH_2_C≡N moieties (since the axial rotation of the nitrile -C≡N group does not lead to a change in molecular dipole moment), has again short relaxation times and low activation energy (E_a_ ≈ 17 kJ mol^−1^). It can be observed, by the comparison of panels (a) and (b) of Figure 2, that, at a given temperature, the relaxation times in these dinitriles are shorter (that is, the conformational dynamics are faster) than those of the haloalkanes, presumably due to the significantly lower mass of the nitrile group. Interestingly, the comparison of the intramolecular dynamics in both plastic crystalline and supercooled liquid phases (shown for the SN_20_GN_80_ mixture in Figure 2b) makes it clear that the relaxation times are very similar in both liquid and plastic crystal phases [12]. In plastic crystals mixtures with a majority of SN molecules, the intramolecular relaxation time is shorter (i.e., the dynamics are faster) [13], as may be expected from the lower relative mass of succinonitrile compared with glutaronitrile (see the insets to Figure 2b).

### 3.2. Compounds with Ketone–Benzene Cross-Conjugation

Increasing the scale of chemical complexity, we now consider compounds containing a carbonyl (ketone) group adjacent to an aromatic (benzene) ring. Figure 3a displays the conformational relaxation times observed in three such aromatic ketones, namely, acetophenone, 1,4-diacetyl benzene, and 4-acetylbiphenyl [47], dispersed in a polystyrene matrix. The activation energy is very similar for the three compounds, being roughly 30 kJ mol^−1^ in all three. Electronic cross-conjugation effects between the carbonyl and phenyl functional groups tend to stabilise fully planar conformations of the ketobenzene moiety [48,49]. As displayed in Figure 3a, such a cross-conjugation effect, together with the relatively large steric hindrance and moment of inertia associated with the phenyl ring, leads to a significantly higher activation barrier and much longer relaxation times of the aromatic ketones compared with the alkanes with a terminal acetyl group, such as butyl acetate (also shown in Figure 3a for comparison purposes).

While the activation barrier is similar for all three compounds containing a ketophenyl moiety, the relaxation times are clearly different, with the smallest compound (acetophenone) displaying the slowest relaxation dynamics. In fact, there is no simple correlation between the chemical structure and the absolute relaxation time. This is shown in more detail in the inset to Figure 3a, which compares the relaxation times of 2-acetyl naphthalene and of two structural isomers of acetyl phenanthrene (after Ref. [48]) with those of acetophenone and acetyl biphenyl. It can be observed that the naphthalene derivative, which only contains two fused benzene rings, has a higher activation energy and slower dynamics compared to both phenanthrene derivatives, despite the fact that the latter contains three fused benzene rings. It is clear that the details of cross-conjugation are here more relevant than the impact of steric hindrance (excluded volume interactions) and the moment of inertia. This is also testified by the fact that the two phenanthrene derivatives have significantly different relaxation times, despite differing only in the position of the acetyl group with respect to the farthermost fused benzene ring.

Figure 3b collects the relaxation times of three biologically important compounds (vitamin A [50], vitamin E [51], and ibuprofen [52]) containing both a carbonyl group and an aromatic ring or a conjugated chain, but not directly adjacent to one another, so that cross-conjugation effects should be weak or absent. The relaxation times of 1,4-diacetyl benzene are also shown. Visual comparison indicates, perhaps unexpectedly, that the relaxation time of all four compounds is quite similar. In the case of vitamin A and vitamin E acetate, this might be due to the fact that both possess an acetoxy (-O(C=O)CH_3_) group relatively close to a rigid conjugated structure (a methylated alkene chain in vitamin A acetate and a methylated fused benzene ring in vitamin E acetate), which might increase the steric hindrance effect. In the case of ibuprofen, which is an acid, the relatively slow relaxation might instead be the result of self-association.

### 3.3. Saccharides

Although we do not focus on HB systems, the case of mono- and disaccharides is worth a brief mention. These compounds are known to display conformational isomerism and interconformer relaxation dynamics, but the exact origin of the processes observed by various techniques has been controversial. In the case of dielectric spectroscopy, a series of articles, by Kaminski and Paluch and coworkers, compared the intramolecular relaxation of amorphous monosaccharides and disaccharides obtained by supercooling the liquid state, in particular, glucose, maltose, and their peracetylated derivatives [53,54,55,56]. As visible in Figure 4, the relaxation times and activation energies of all four compounds are very similar. In particular, the activation energies are found to be, in kJ mol^−1^, 42 for glucose, 46 for maltose, and 41 for both peracetylglucose and peracetylmaltose. Similar relaxation times and activation energies are observed for sucrose and cellulose, where the relaxation dynamics are only slightly slower [53]. Contrary to most intramolecular relaxation processes, the dielectric strength of this process depends very strongly on temperature (namely, it decreases with increasing *T*), and the shape of the loss feature is asymmetric, and cannot therefore be modelled with a Cole–Cole function (Equation (1)). Rather, a generalisation of this equation, known as the Havriliak–Negami function [57], had to be employed.

The authors of these comparative studies have concluded that the intramolecular relaxation of glucose and maltose compounds originates from the rotation of the exocyclic –CH_2_OH group in the case of the pristine compounds, and of the acetoxy (-O(C=O)CH_3_) moiety in the case of the peracetylated derivatives [55]. It is, however, worth mentioning that the same authors performed ab initio (DFT) calculations on D-glucose [53], and reported a calculated energy of hydroxymethyl group reorientation of only 15 kJ mol^−1^, whereas the chair-to-boat interconformer transformation of the same molecule has a theoretical activation barrier of 49 kJ mol^−1^, which is much closer to the experimental activation energy of the intramolecular dielectric process. Given that the same intramolecular relaxation is observed in several saccharides, and that it is virtually unaffected by acetylation, another possible rationalisation of this relaxation mode is a boat-to-chair interconformer transition (possibly accompanied by simultaneous reorientation of exocyclic groups to lower the effective energy barrier).

### 3.4. Compounds with Aromatic Halobenzene Rings

Substituted and unsubstituted benzene cycles, especially when they are terminal rings of a larger compound, can display several possible orientations with respect to the rest of the molecule that contains them. The corresponding rotational barriers are usually easy to overcome at room temperature. In order to be visible in dielectric spectroscopy, the phenyl group must have at least one polar substituent, which, in the case of singly-substituted aryl rings, must be located at a position other than position four, as otherwise the rotation of the ring around its external link (at position one) does not lead to a variation in the molecular dipole moment.

A previous study by some of us [58] has shown that some compounds containing halogenated aryl rings, in particular, chlorobenzene and bromobenzene rings, display very similar conformational relaxation times. Here, we extend the comparison to other related compounds. Figure 5 shows the intramolecular relaxation time of two halogenated benzophenone derivatives (*o*-bromobenzophenone and *o,p*’-dichlorobenzophenone) and of two active pharmaceutical ingredients (*o,p*’-dichlorodiphenyldichloroethane, known as mitotane, and 2-[(2,4-dichlorophenyl)methyl]-4-(2,4,4-trimethylpentan-2-yl)phenol, known as clofoctol). It may be observed that the activation energies of the interconformer relaxation is very similar in all four compounds, being 25 ± 1 kJ mol^−1^ for mitotane, 21 ± 2 kJ mol^−1^ for *o,p*’-dichlorobenzophenone, 28 ± 1 kJ mol^−1^ for *o*-bromobenzophenone [58], and 25.6 ± 0.1 kJ mol^−1^ for clofoctol [59], respectively. The relaxation times are virtually identical for the three compounds containing a chlorobenzene ring, while they are only slightly longer for the compound containing a bromobenzene ring, as may be expected considering the significantly large bromine mass (Br has a mass larger than the whole unsubstituted phenyl ring, and more than twice the mass of Cl), which leads to a larger moment of inertia of the bromobenzene ring compared to the chlorobenzene one.

For mitotane and *o,p*’-dichlorobenzophenone, both of which possess two chlorobenzene rings, only the rotation of one of them is active in dielectric spectroscopy, being the other Cl substituent at position four. Other than the intramolecular one depicted in Figure 5, clofoctol only displays a structural (α) and a JG (*β*) relaxation. This suggests that the phenol group of clofoctol does not give rise to a separate relaxation feature, which may be rationalised, since it is covalently linked on either side to two relatively long molecular subunits, and it is also likely involved in H-bonding in condensed amorphous phases.

It is also interesting to note that the relaxation times of the active pharmaceutical ingredients are almost identical to those of *o,p*’-dichlorobenzophenone, despite the fact that, in the latter case, the dielectrically active chlorobenzene ring is attached to a ketone group next to another aromatic ring, leading to cross-conjugation effects [36,60] of the same type as those mentioned in Section 3.2 [48,49] (the pharmaceutical compounds are instead not ketone derivatives). This shows that the relaxation times (and activation energies) of conformational interconversion processes involving the rotation of halogenated aromatic rings may be relatively insensitive to the details of the molecular electronic configuration (this is not true for compounds containing nonfunctionalised phenyl rings [61]).

It is interesting to remark that biclotymol, another pharmaceutically active compound possessing two geminal aryl rings, but which are functionalised also by methyl, isopropyl, and hydroxyl groups, displays much slower relaxation dynamics (by more than two orders of magnitude) than clofoctol and mitotane, and significantly higher activation energy (75.5 ± 0.7 kJ mol^−1^) [62]. These effects are likely due both to the larger mass and steric hindrance of the conjugated rings of the biclotymol compound, and the formation of intermolecular HBs via the hydroxyl groups [63].

### 3.5. Compounds with Flexible Rings

A recent article by some of us [64] focused on the intramolecular relaxations of three benzodiazepine derivatives (diazepam, nordazepam, and tetrazepam), whose molecular structures are shown as insets in Figure 6a. Benzodiazepines constitute a large class of commercial pharmaceutical compounds, possessing a common molecular structure that consists of a rigid benzene ring fused to a flexible seven-member diazepine ring. The benzodiazepines reported here also display a third six-membered ring, covalently attached via a single bond to the diazepine ring. This third cycle, which is either a benzene or a cyclohexane ring, possesses at least one torsional degree of freedom, but it is not dielectrically active due to the lack of a dipole moment. All three benzodiazepine compounds could be vitrified by cooling the liquid phase down to the glass state, where they all also displayed, besides a JG relaxation, an intramolecular relaxation, whose characteristic times are shown as Arrhenius plots in Figure 6a.

The only internal degree of freedom of these derivatives that is dielectrically active is the chirality inversion between P and M conformers, schematically displayed in Figure 6b, which is common to all three. This degree of freedom of the benzodiazepine compounds is activated both in the gas phase and in solution, where they display interconversion dynamics between the P and M conformations accompanied by a reorientation by 60° of the -CH_2_ moiety attached to the carbonyl group [65]. Both conformers are, moreover, present in the crystal phase of all three compounds [64].

As visible in panel (a), this conformational relaxation has, at a given fixed temperature, very similar relaxation times in all three compounds. The activation energies obtained from dielectric experiments are reported in Table 1, together with those available from NMR studies and ab initio calculations [66,67]. Remarkable agreement is observed, despite the fact that the probed phases are different in all three methods: the dielectric measurements were performed in the amorphous solid state of the pure compounds, the NMR data were acquired on solutions, and the ab initio calculations refer to the gas phase. It should also be noted that the relaxation times are only available from the dielectric spectroscopy experiments, which constitutes an obvious advantage of this technique compared with temperature-dependent NMR or ab initio calculations, which only yield activation barriers.

Only a few other dielectric studies exist on polycyclic compounds containing flexible ring structures. In all cases, the observed intramolecular dynamics are faster than in the benzodiazepine case. In the case of vitamin E acetate, a compound already encountered in Section 3.2, the authors of the same dielectric study [51] report another secondary relaxation besides the one associated with the acetoxy group. The authors tentatively assign this relaxation, which is slower than the one associated with the acetoxy group, to a JG relaxation, despite the fact that (as they show in Figure 5 of their article) the predictions of the coupling model [4,5] for the primitive relaxation times are off by one-and-a-half decades. It is quite possible that the observed secondary relaxation stems instead from the internal dynamics of the tetrahydropyran (oxane) ring fused to the aromatic ring (see the molecular structure in Figure 3b), which would be dielectrically active thanks to the oxygen substituent in the oxane ring.

Two recent studies [68,69] reported the conformational relaxation dynamics of “thiacrown” heterocyclic compounds. We review here the dynamics of compounds containing either a dithia-cyclohexane ring, a dithiaoxa-cyclononane, or a tetrathiacyclododecane ring, fused to a methylbenzene ring (see the insets to Figure 6a). The relaxation times and activation energies of these processes were much smaller than those for benzodiazepines. The relaxation of the smallest compound (containing two six-member rings) is faster than that of the other thiacrown compounds and has the smallest energy barrier (10 kJ mol^−1^). DFT calculations carried out in the original work [68] indicate that this relaxation likely corresponds to the conversion between two isoenergetic half-chair conformations of the dithia-cyclohexane ring, which occurs via an intermediate high-energy state. The activation energy of the ring-buckling relaxation is the same (19 kJ mol^−1^) for the larger thiacrown compounds, and the dynamics at a given temperature are slightly faster for the 12-member thiacrown than for the 9-member thiacrown. These similarities suggest that the conformational change occurring in both compounds (TC2 and TC3 in Figure 6a) involves the same moiety (perhaps a –CH_2_CH_2_– moiety linked to consecutive S or O atoms). Apart from this, no clear correlation can be devised between the relaxation time and activation barrier on one side, and the molecular structure or composition on the other (for example, it might be expected that there is a less stringent geometric constraint of the cyclic topology with a larger number of atoms in the ring, but this is not systematically observed).

**Figure 6 ijms-24-17189-f006:**
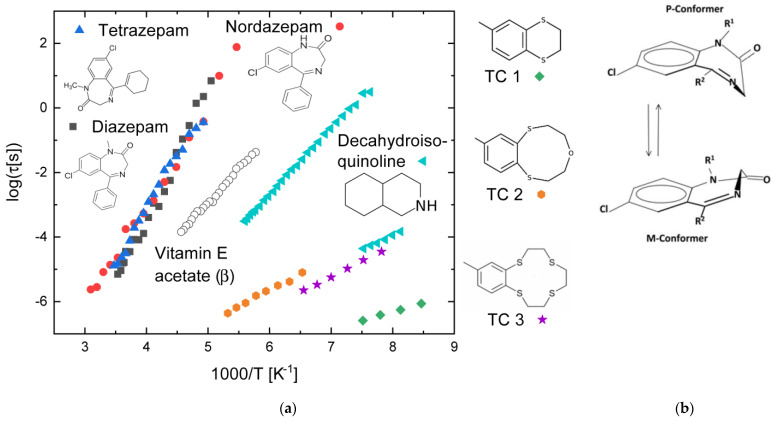
(**a**) Characteristic relaxation times of the diazepine ring inversion in three benzodiazepine derivatives (diazepam, nordazepam, and tetrazepam, after Ref. [64]), of the interconformer conversion in the double-ring decahydroisoquinoline compound [70], and of the ring-buckling relaxation time in three thiacrown heterocycles (labelled as TC in the panel legend) [68,69]. The relaxation times of the β relaxation of vitamin E acetate (open circles) are also shown for comparison [51]. (**b**) Schematic representation of the two possible (M- and P-) conformers of the fused benzodiazepinic ring of opposite chirality. The ring-inversion relaxation interchanges these two conformations.

In Figure 6a, the relaxation times of thiacrown compounds and those of vitamin E acetate are also compared with those reported for the decahydroisoquinoline compound, which exhibits two intramolecular relaxations whose interpretation is controversial [70,71]. The activation energy of the faster relaxation is 17 kJ mol^−1^, close to that of the larger thiacrowns, while that of the slower relaxation is 40 kJ mol^−1^, virtually identical to that of the slower relaxation of vitamin E acetate [51], but the relaxation times at a given temperature differ by almost a factor of 100. The precise value of the activation barriers and of relaxation times depends on the allowed conformations, which are different depending on the number and types of atoms present in the ring, and, in the case of decahydroisoquinoline, may also depend on formation of HBs. Hence, no simple structure–property relation can be expected to hold, and the coincidence of activation energies and the similarity of relaxation times for the conformational relaxations of cyclic molecules may be accidental.

### 3.6. Polyesters and Polyesteramides

We end with a short subsection on synthetic polyesters. The conformational relaxation of poly(ε-caprolactone) has been studied by dielectric spectroscopy by at least two groups [72,73], which found an activation energy of 39 ± 2 kJ mol^−1^ for this process (see Figure 7). Recently, some of us [74] employed dielectric spectroscopy to study the relaxation behaviour of PADAS, a polyesteramide based on alanine, dodecanediol, and sebacic acid. As visible in Figure 7, the conformational relaxation of PADAS has a very similar relaxation time and basically the same activation energy as the local secondary relaxation dynamics of polycaprolactone (the activation energy was 39 ± 1 kJ mol^−1^ in the PADAS case). Both polycaprolactone and PADAS possess ester groups, which is the only polar group present in the former. Hence, the secondary dynamics observed in these polymers must stem from local configurational reorientation of the ester group, likely involving a torsional rotation around a backbone C−O bond that modifies the spatial orientation of the carbonyl group [74].

The observation of very similar relaxation behaviour in long macromolecules of different molecular weight and chemical composition indicates that such conformational relaxations are local processes, in the sense that they involve only a few polar groups (one or two monomer units) and do not involve directly the rest of the polymer chain (the activation energy would be much higher in the latter case). It is worth recalling from Section 3.3 that some polysaccharides also exhibit secondary relaxations that can be correlated with the conformational dynamics of single (glucose) monomers [53].

Interestingly, polyesters with a smaller monomer unit, such as polylactic acid or poly(glycolide), were not reported to display conformational relaxations in dielectric studies [75,76]. This suggests that the observation of a well-defined intramolecular relaxation is hindered if the polar group involved is not sufficiently diluted along the polymer chain. The results reviewed in this subsection indicate that distinct synthetic polymers may display a similar, if not equivalent, local conformational relaxation, provided that the polar groups involved are sufficiently spaced along the backbone (that is, provided they are separated by relatively long aliphatic carbohydrate subchains). It should be interesting to test this conjecture with other families of synthetic polymers.

## 4. Discussion

We have shown in the previous sections that, in some cases, it is possible to assign the observed conformational relaxation (mainly) to the rotation of a single functional group. This is visualised in Figure 8, which summarises in the same Arrhenius plot the data for a number of compounds discussed in this review (see the figure legend). Figure 8 confirms the early observation of a correlation of the relaxation times and activation energy with the size of the molecular moiety involved in the relaxation process [11]. It may also be discerned that, in the limit of very high temperature (1/T→0), the relaxation times of all processes considered tend to a value of the order of 10^−14^ s, which thus represents the typical value for the parameter $\tau_{\infty }$ in the Arrhenius Equation (2) for intramolecular processes (see Section 2). The value of 10^−14^ s corresponds to the typical timescale of molecular vibrational modes.

The results reviewed in Section 3 show that the time is ripe for the construction of an experimental database of characteristic times and activation barriers for intramolecular processes of flexible organic molecules in disordered condensed (liquid and glass) phases. Such a database would provide useful information about the intermolecular steric interactions in condensed phases, which may also be of biological relevance, as well as important insights on phase behaviour, (bio)chemical activity, and the reliability of molecular dynamics simulations of the compounds of interest. From the experimental point of view, in fact, the occurrence of dynamic conformational isomerism has consequences for the effective molecular shape and steric intermolecular interactions in condensed phases, leading to the possible formation of liquid crystalline or plastic crystalline phases, as mentioned in Section 1. From the computational point of view, a database of conformational relaxation times and activation energies may be helpful to test the validity of the torsional force fields employed in molecular dynamics simulations of condensed phases (liquids, orientationally disordered solids) of flexible molecules [15,16,17,18,19,20].

From a more applied perspective, knowledge of conformational relaxations taking place in the amorphous solid state can provide important information concerning the kinetic stability of molecular glasses. Especially in the case of pharmaceutical compounds, knowledge of the molecular relaxation dynamics may be also important to understand the mechanism of action of drugs in vivo, as the conformational disorder of a molecule in aqueous solution implies a more dynamic character of the surrounding hydration shell, which can affect its interaction with other (bio)molecules, and also because molecular flexibility may facilitate the induced-fit interaction between the drug and the active site of its target [77,78,79].

The construction of a database would allow a better determination of the validity of the idea that intramolecular relaxation times can be common to distinct molecules carrying the same or similar functional group, and would permit to test and better quantify the possible dependence of conformational dynamics on the molecular environment. The results reviewed in Section 3.6 show that such a database can also be extended to the case of macromolecular (polymer) samples. In the case of nonself-assembling compounds, it would be interesting to carry out a comparative analysis of computational (ab initio) studies to understand under which circumstances the activation barrier for internal rotation is independent of the details of the (rest of the) molecular structure, and what parameters determine in such cases the activation energy and absolute relaxation time of the interconformer relaxation processes.

HB interactions are likely to affect the relaxation dynamics of the HB donor and acceptor functional groups to an extent determined by the density of bonds [23]. Disentangling the effect of the HBs on the conformational dynamics of the condensed amorphous phases of organic molecules is an interesting line of research to be pursued by carrying out systematic studies involving the comparison between the gas-phase and condensed-phase properties of HB compounds.

## 5. Conclusions

We presented a review of intramolecular conformational relaxations in the supercooled liquid and glassy states of polar small-molecule glass-formers, and a few polymeric ones, as observed by means of dielectric spectroscopy. The data reviewed suggest that, in the case of compounds containing relatively rigid halophenyl and chlorophenyl rings, aromatic ketophenyl moieties, or terminal halomethane, acetyl, and nitrile groups, the relaxation dynamics display characteristic times and activation energies that are similar for compounds that have the same ring structure or functional group. For these compounds, the activation energy, as well as the characteristic interconformer relaxation time at a given fixed temperature, increase with the increasing size (mass, moment of inertia) of the moiety involved in the conformational dynamics. Due to this correlation, when plotted in the same Arrhenius plot, the relaxation times of these compounds tend, in the limit of very high temperature, to the typical vibration timescale of 10^−14^ s. For some families of cyclic molecules, such as diazepine, glucose, and maltose, the relaxation times cluster together when plotted in the same Arrhenius plot; for others, such as thiacrown compounds and six-membered heterocycles containing O or N atoms, the relaxation timescale and activation energies are observed in distinct ranges, and no simple structure–property correlation is observed. Finally, in some cases, the conformational relaxation rates and activation energies are similar for polymeric systems containing the same functional groups of rings, such as polysaccharides and polyesters. Further research is needed to test the range of validity of these observed similarities in the amorphous phase of similar molecular and macromolecular compounds.

## Figures and Tables

**Figure 1 ijms-24-17189-f001:**
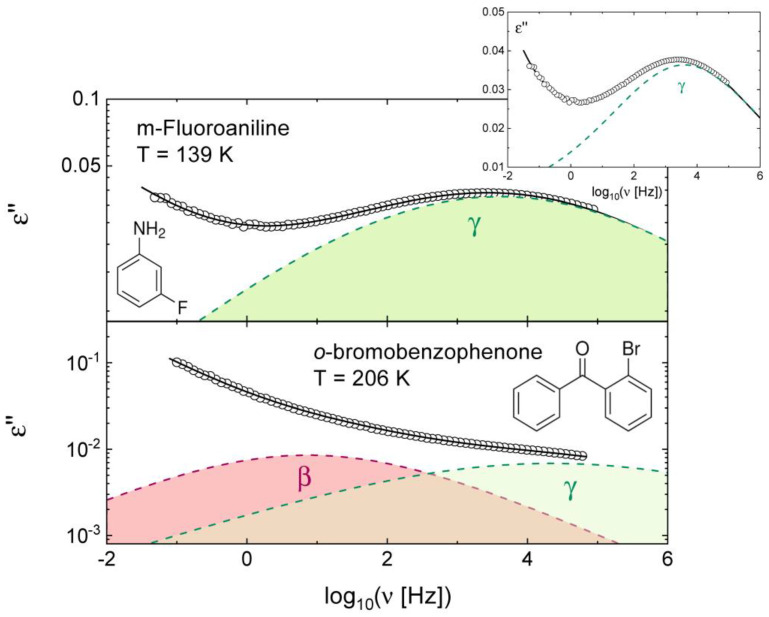
Isothermal dielectric loss spectra of glassy *m*-fluoroaniline (**upper panel**) and *o*-bromobenzophenone (**lower panel**) in double logarithmic scale. The molecular structures of both compounds are shown as insets. Both spectra were measured in parallel-plate capacitor geometry on samples obtained by supercooling the pure liquids down to the indicated temperatures, which are below the glass transition temperature of either compound. Open markers are experimental points, continuous lines are fits, and dashed lines and shaded areas represent fit components, each modelled as the imaginary part of Equation (1) (see the text for details). The γ component corresponds to a conformational relaxation, while the β component, observed only in the lower panel (at lower frequency), is a Johari–Goldstein relaxation. The inset to the upper panel displays the *m*-fluoroaniline spectrum employing a linear (rather than logarithmic) vertical scale.

**Figure 2 ijms-24-17189-f002:**
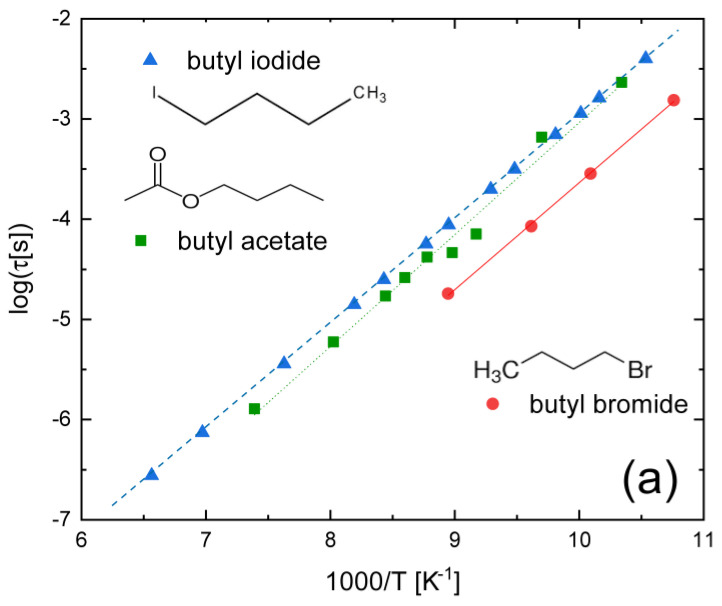
(**a**) Conformational relaxation times of three butane derivatives with a polar terminal group: butyl iodide, butyl bromide [11], and butyl acetate [41]; (**b**) Conformational relaxation times of dinitrile mixtures composed of succinonitrile and glutaronitrile in the supercooled liquid phase with a high glutaronitrile content [12], and in the plastic crystalline phase with a high glutaronitrile [12] or high succinonitrile content [13]. The legend indicates the molar composition of the samples.

**Figure 3 ijms-24-17189-f003:**
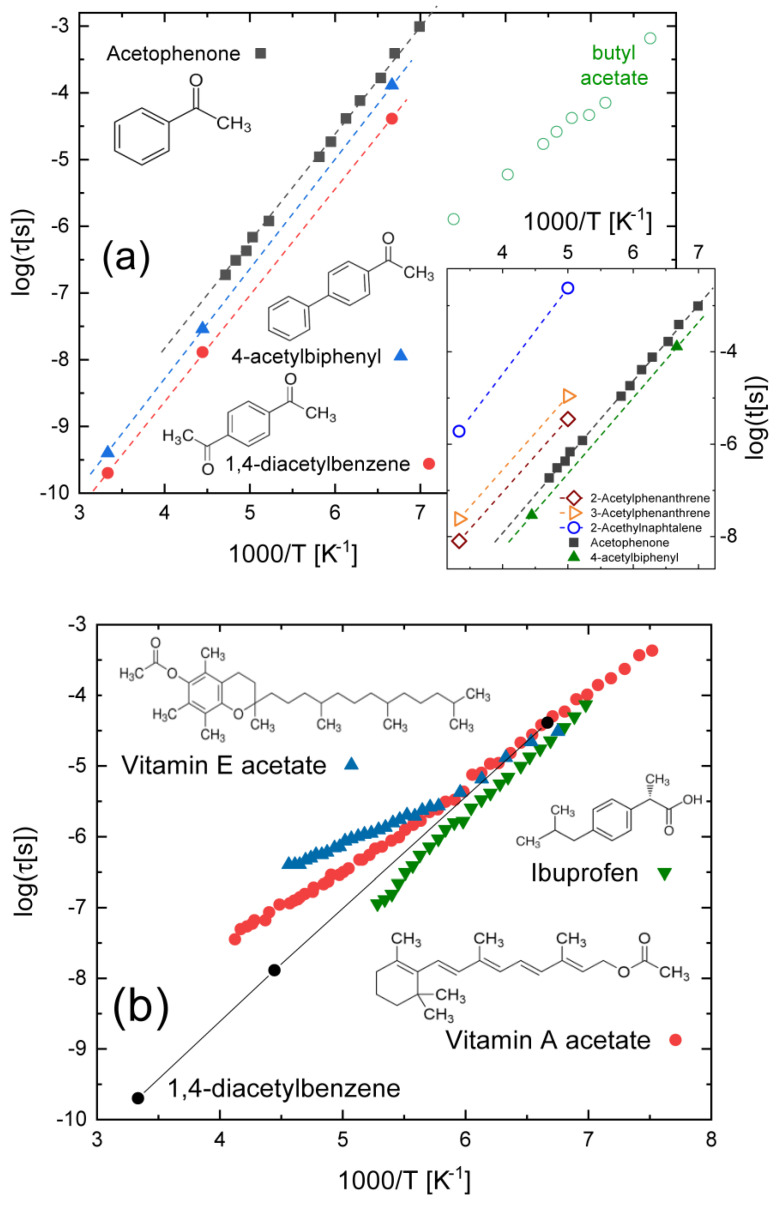
(**a**) Molecular structures and intramolecular relaxation times of acetophenone, 4-acetylbiphenyl, and 1,4-diacetyl benzene dispersed in polystyrene (after Ref. [47]). The relaxation time corresponding to the conformational relaxation in butyl acetate is also shown for comparison. Inset to (**a**): Conformational relaxation times of acetyl naphthalene and of two acetyl-phenanthrene structural isomers [48], as measured on dilute solutions in p-xylene, compared with those of acetophenone and 4-acetylbiphenyl of the main panel; (**b**) Molecular structures and intramolecular relaxation times of supercooled liquid vitamin A acetate [50], vitamin E acetate [51], and ibuprofen [52]. The relaxation times of 1,4-diacetyl benzene (data of panel (**a**)) are also shown for comparison.

**Figure 4 ijms-24-17189-f004:**
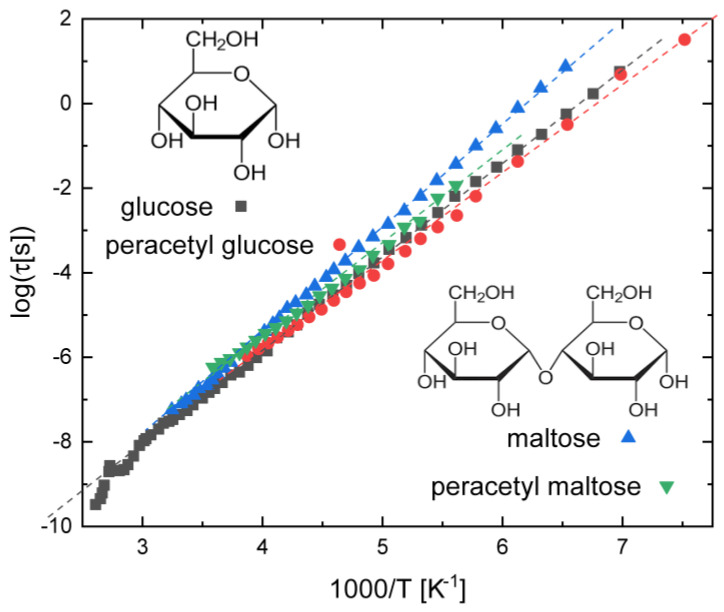
Intramolecular relaxation times of the glucose monosaccharide and of the maltose disaccharide, and of their peracetylated derivatives (octacetyl glucose and pentacetyl maltose); after Ref. [55].

**Figure 5 ijms-24-17189-f005:**
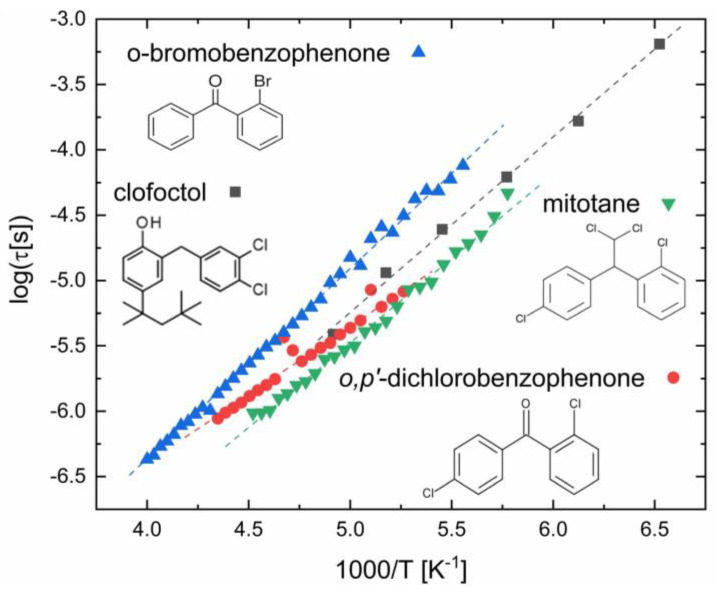
Interconformer relaxation times of clofoctol [59], mitotane, chlorobenzophenone, and bromobenzophenone (after Ref. [58]). The latter compound contains a brominated phenyl ring and displays slower dynamics, as might be expected from its larger mass and moment of inertia.

**Figure 7 ijms-24-17189-f007:**
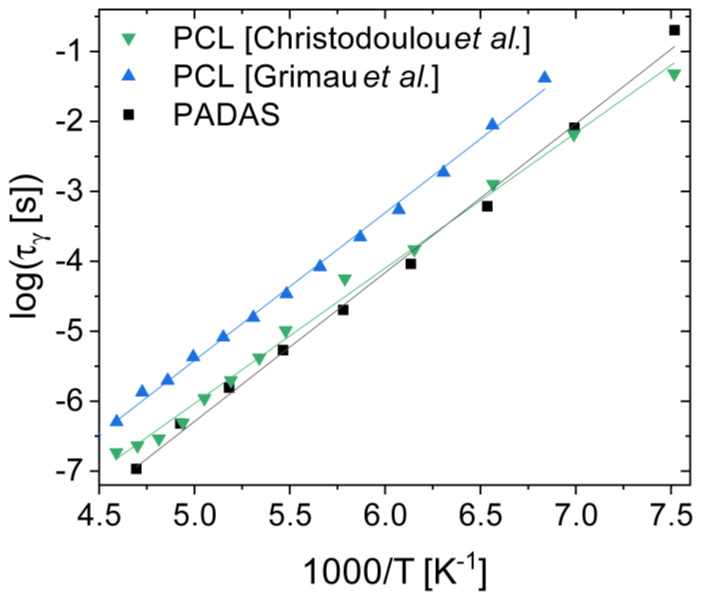
Secondary relaxation times of a polyesteramide, PADAS (data of Ref. [74]), and of two samples of poly(ε-caprolactone) (PCL) (Refs. [72,73]).

**Figure 8 ijms-24-17189-f008:**
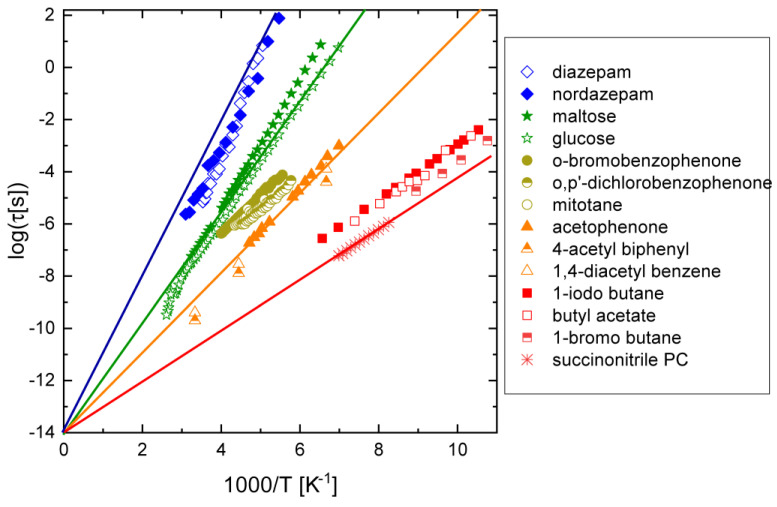
Arrhenius plot of the relaxation times (markers) of selected dielectric processes that can be ascribed to the rotation or flexing of (relatively rigid) rings and terminal groups (diazepine, glucose, bromophenyl, and chlorophenyl rings; aromatic ketophenyl moieties; terminal iodomethane, bromomethane, acetyl, and nitrile groups). Continuous lines are guides to the eye that reach a value of the relaxation time of 10^−14^ s at infinite temperature (1000/*T* → 0).

**Table 1 ijms-24-17189-t001:** Activation energy *E_a_*, in kJ mol^−1^, of the diazepine ring-inversion relaxation of three benzodiazepine derivatives.

Compound	Dielectric *E_a_*	NMR *E_a_*	Ab Initio *E_a_*
Diazepam	72 ± 9	74 ^1^	72.4–74.1 ^2^
Nordazepam	58 ± 5	52 ^1^	44.8–47.3 ^2^
Tetrazepam	58 ± 5	–	

^1^ After Ref. [66]; ^2^ After Ref. [67].

## Data Availability

Not applicable.

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
