# Peer review of "Dielectric Spectroscopy Studies of Conformational Relaxation Dynamics in Molecular Glass-Forming Liquids"

_ijms, 2023, doi:10.3390/ijms242417189_

Round 1

Reviewer 1 Report

Comments and Suggestions for Authors

The authors present an interesting overview of the current scientific achievements devoted to study of relaxation times and activation energies of intramolecular, conformational relaxation processes in small-molecule glass formers. As a method of study the broadband dielectric spectroscopy was chosen.

The subject matter taken by the authors of the paper can be considered up-to-date and important as far as fundamental microscopic understanding of the properties of amorphous solids, in particular their ageing and their glass transition behaviour, requires knowledge of the temperature dependence of microscopic relaxation processes.

The paper is interesting and written in good English. The title is concise and the abstract is a precise summary of the contents. The literature study is comprehensive.

Scientific quality of the review paper is good with a high "impact" to the field. I recommend the papre for publication after minor, technical improvements. I suggest that the Authors check if the Equation 2 is correct (minus sign in the exp.factor is missing).

I would also welcome an additional information in Section 2 (Method) about the equipment used for particular studies. The point is that dielectric spectroscopy is very sensitive to external disturbances. Moreover, measurements carried out within the frequency range higher than 1MHz are subjected to instrumental errors (even in the case of high quality measuring equipment) - Fig.3 contains data with relaxation frequency higher than 10MHz (black squares). If possible, please give information how the relaxation time was derived from experimental data. One or two example plots of dielectric permittivity spectroscopic dependence will be welcome.

Reviewer 2 Report

Comments and Suggestions for Authors

After studying the considered paper, I give the following statement. The results summarized in this paper show that it is time to build an experimental database of characteristic times and activation barriers for intramolecular processes in flexible organic molecules. Such a database would be useful. The overview given in the paper is structured into successive subsections focusing on different types of functional groups that can contribute to the dynamic conformational isomerism of organic compounds. The authors start from smaller functional groups (terminal halomethyl, nitrile, acetyl, and acetoxy groups) to move to rigid or flexible ring groups (aromatic halobenzenes, saccharide rings, benzodiazepines, thiacrown ethers), ending with a subsection on polyesters.

Notes: the equations are given in the correct form. The pictures adequately illustrate the described issue. The contribution will certainly benefit a wide range of interested parties. A stronger emphasis on the scientific contribution is needed, which is missing in the article. After editing and supplementing the article with the above, the article can be published.

Reviewer 3 Report

Comments and Suggestions for Authors

Authors of this study have reviewed the current literature highlighting usefulness of broadband dielectric spectroscopy of molecular glass-forming liquids.

1.     Authors have stated that strong intermolecular interactions such as hydrogen (H) bonding may hinder or slow down intramolecular relaxations. Authors should be aware that hydrogen bonding should be symbolically represented by (HB), and the IUPCAC paper that provides a semi-official definition to it should be cited. Moreover, I am not sure how intermolecular interactions represent an obstacle for intramolecular interactions. Authors should vivify this with appropriate examples and citations.

2.     Why is the text in the results and discussion (pages 4 and 5) is shown in bold font?

3.     This study has overlooked to review to compounds that do not exhibit (extensive) H-bonding, this narrowing the usefulness of the reviewer for vast research audience. I suggest additions during revision.

4.     In-depth interpretations are a requisite to enhance our current understanding of the linear relationships shown in graphs in Figs. 1-6. Authors should provide their views rather than just assembling results from other studies.

5.     The conclusion of the paper is not very compact and comprehensive. It also includes a Fig, Fig. 7, which should not be. I urge authors to rewrite this section and move Fig. 7 to the discussion section of the ms.

6.     The useful of dielectric spectroscopy in revealing the glass transition temperature and elastic constants of materials (both thermosetting and thermoplastics) should be incorporated.

Comments on the Quality of English Language

--

Round 2

Reviewer 3 Report

Comments and Suggestions for Authors

I am fine with the corrections made by the authors, who have also addressed my suggestions to improve their ms. Revisions made are highlighted in the ms. I have no problems if you this ms is accepted for publication in IJMS.

Comments on the Quality of English Language

--